# Sleep Duration is Inversely Associated with Serum Uric Acid Concentrations and Uric Acid to Creatinine Ratio in an Elderly Mediterranean Population at High Cardiovascular Risk

**DOI:** 10.3390/nu11040761

**Published:** 2019-04-01

**Authors:** Christopher Papandreou, Nancy Babio, Andrés Díaz-López, Miguel Á. Martínez-González, Nerea Becerra-Tomas, Dolores Corella, Helmut Schröder, Dora Romaguera, Jesús Vioque, Ángel M. Alonso-Gómez, Julia Wärnberg, Alfredo J. Martínez, Lluís Serra-Majem, Ramon Estruch, Araceli Muñoz-Garach, José Lapetra, Xavier Pintó, Josep A. Tur, Antonio Garcia-Rios, Aurora Bueno-Cavanillas, Miguel Delgado-Rodríguez, Pilar Matía-Martín, Lidia Daimiel, Vicente Martín-Sánchez, Josep Vidal, Clotilde Vázquez, Emilio Ros, Miguel Ruiz-Canela, Mónica Bulló, Jose V. Sorli, Mireia Quifer, Antoni Colom, Alejandro Oncina-Canovas, Lucas Tojal-Sierra, Javier Barón-López, Napoleón Pérez-Farinós, Itziar Abete, Almudena Sanchez-Villegas, Rosa Casas, José C. Fernández-Garcia, José M. Santos-Lozano, Emili Corbella, Maria del M. Bibiloni, Javier Diez-Espino, Eva M. Asensio, Laura Torras, Marga Morey, Laura Compañ-Gabucio, Itziar Salaverria-Lete, Juan C. Cenoz-Osinaga, Olga Castañer, Jordi Salas-Salvadó

**Affiliations:** 1Universitat Rovira i Virgili, Departament de Bioquímica i Biotecnologia, Unitat de Nutrició, 43201 Reus, Spain; papchris10@gmail.com (C.P.); nancy.babio@urv.cat (N.B.); andres.diaz@urv.cat (A.D.-L.); nerea.becerra@urv.cat (N.B.-T.); monica.bullo@urv.cat (M.B.); 2Institut d’Investigació Sanitària Pere Virgili (IISPV), 43007 Reus, Spain; 3Centro de Investigación Biomédica en Red Fisiopatologia de la Obesidad y la Nutrición (CIBEROBN), InstituteofHealth Carlos III, 28029 Madrid, Spain; mamartinez@unav.es (M.Á.M.-G.); dolores.corella@uv.es (D.C.); mariaadoracion.romaguera@ssib.es (D.R.); angelmago13@gmail.com (Á.M.A.-G.); jwarnberg@uma.es (J.W.); jalfmtz@unav.es (A.J.M.); lluis.serra@ulpgc.es (L.S.-M.); restruch@clinic.ub.es (R.E.); aracelimugar@gmail.com (A.M.-G.); jlapetra@ono.com (J.L.); xpinto@bellvitgehospital.cat (X.P.); pep.tur@uib.es (J.A.T.); angarios2004@yahoo.es (A.G.-R.); clotilde.vazquez@fjd.es (C.V.); eros@clinic.ub.es (E.R.); mcanela@unav.es (M.R.-C.); Jose.Sorli@uv.es (J.V.S.); mire_mk@hotmail.com (M.Q.); antonicolom@gmail.com (A.C.); lutojal@hotmail.com (L.T.-S.); fjbaron@gmail.com (J.B.-L.); napoleon.perez@uma.es (N.P.-F.); iabetego@unav.es (I.A.); almudena.sanchez@ulpgc.es (A.S.-V.); rcasas1@clinic.cat (R.C.); josecarlosfdezgarcia@hotmail.com (J.C.F.-G.); jsantos11@us.es (J.M.S.-L.); xcorbella@csub.scs.es (E.C.); mar.bibiloni@uib.es (M.d.M.B.); javierdiezesp@ono.com (J.D.-E.); eva.asensio.marquez@gmail.com (E.M.A.); lauratorresmota@gmail.com (L.T.); marga.morey@yahoo.es (M.M.); ITZIAR.SALAVERRIALETE@osakidetza.eus (I.S.-L.); jc.cenoz.osinaga@cfnavarra.es (J.C.C.-O.); ocastaner@imim.es (O.C.); 4University Hospital of Sant Joan de Reus, Nutrition Unit, 43204 Reus, Spain; 5Department of Preventive Medicine and Public Health, University of Navarra, IDISNA, 31009 Pamplona, Spain; 6Department of Nutrition, Harvard T.H. Chan School of Public Health, Boston, MA 02115, USA; 7Department of Preventive Medicine, University of Valencia, 46010 Valencia, Spain; 8Cardiovascular Risk and Nutrition research group (CARIN), Hospital del Mar Research Institute (IMIM), 08003 Barcelona, Spain; HSchoeder@imim.es; 9Clinical Epidemiology and Public Health Department, Health Research Institute of the Balearic Islands (IdISBa), 07120 Palma de Mallorca, Spain; 10CIBER de Epidemiología y Salud Pública (CIBERESP), Instituto de Salud Carlos III, 28009 Madrid, Spain; vioque@umh.es (J.V.); abueno@ugr.es (A.B.-C.); mdelgado@ujaen.es (M.D.-R.); vicente.martin@unileon.es (V.M.-S.); aoncina@umh.es (A.O.-C.); lcompan@umh.es (L.C.-G.); 11Miguel Hernandez University, ISABIAL-FISABIO, 03202 Alicante, Spain; 12Department of Cardiology, Organización Sanitaria Integrada (OSI) ARABA, University Hospital Araba, 01009 Vitoria-Gasteiz, Spain; 13Department of Nursing, School of Health Sciences, University of Málaga-IBIMA, 29016 Málaga, Spain; 14Department of Nutrition, Food Science and Physiology, University of Navarra, IDISNA, 43204 Pamplona, Spain; 15University of Las Palmas de Gran Canaria, Research Institute of Biomedical and Health Sciences (IUIBS), Preventive Medicine Service, Centro Hospitalario Universitario Insular Materno Infantil (CHUIMI), Canarian Health Service, 35001 Las Palmas, Spain; 16Department of Internal Medicine, IDIBAPS, Hospital Clinic, University of Barcelona, 08007 Barcelona, Spain; 17Virgen de la Victoria Hospital, Department of Endocrinology (IBIMA), University of Málaga, 29016 Málaga, Spain; 18Department of Family Medicine, Research Unit, Distrito Sanitario Atención Primaria Sevilla, 41013 Sevilla, Spain; 19Lipids and Vascular Risk Unit, Internal Medicine, Hospital Universitario de Bellvitge, Hospitalet de Llobregat, 08907 Barcelona, Spain; 20Research Group on Community Nutrition & Oxidative Stress, University of Balearic Islands & Health Research Institute of the Balearic Islands (IdISBa), 07120 Palma de Mallorca, Spain; 21Department of Internal Medicine, Maimonides Biomedical Research Institute of Cordoba (IMIBIC), Reina Sofia University Hospital, University of Cordoba, 14071 Cordoba, Spain; 22Department of Preventive Medicine, University of Granada, 18071 Granada, Spain; 23Division of Preventive Medicine, Faculty of Medicine, University of Jaén, 23071 Jaén, Spain; 24Department of Endocrinology and Nutrition, Instituto de Investigación Sanitaria Hospital Clínico San Carlos (IdISSC), 28040 Madrid, Spain; mmatia@ucm.es; 25Nutritional Genomics and Epigenomics Group, IMDEA Food, CEI UAM + CSIC, 28049 Madrid, Spain; lidia.daimiel@imdea.org; 26Institute of Biomedicine (IBIOMED), University of León, 24071 León, Spain; 27CIBER Diabetes y enfermedades Metabólicos (CIBERDEM), Instituto de Salud Carlos III (ISCIII), 28029 Madrid, Spain; JOVIDAL@clinic.ub.es; 28Endocrinology and Nutrition Department, Hospital Clinic Universitari, Barcelona, Spain; Institut d’Investigacions Biomèdiques August Pi Sunyer (IDIBAPS), 08036 Barcelona, Spain; 29Department of Endocrinology and Nutrition, University Hospital Fundación Jimenez Díaz, 28040 Madrid, Spain; 30Lipid Clinic, Endocrinology and Nutrition Service, Institut d’Investigacions Biomediques August Pi Sunyer (IDIBAPS), Hospital Clinic, University of Barcelona, 08007 Barcelona, Spain; 31Department of Public Health, University of Málaga-IBIMA, 29016 Málaga, Spain

**Keywords:** sleep duration, actigraphy, serum uric acid, serum uric acid to creatinine ratio

## Abstract

The aim of the study was to evaluate sleep duration and sleep variability in relation to serum uric acid (SUA) concentrations and SUA to creatinine ratio. This is a cross-sectional analysis of baseline data from 1842 elderly participants with overweight/obesity and metabolic syndrome in the (Prevención con Dieta Mediterránea) PREDIMED-Plus trial. Accelerometry-derived sleep duration and sleep variability were measured. Linear regression models were fitted to examine the aforementioned associations. A 1 hour/night increment in sleep duration was inversely associated with SUA concentrations (β = −0.07, *p* = 0.047). Further adjustment for leukocytes attenuated this association (*p* = 0.050). Each 1-hour increment in sleep duration was inversely associated with SUA to creatinine ratio (β = −0.15, *p* = 0.001). The findings of this study suggest that longer sleep duration is associated with lower SUA concentrations and lower SUA to creatinine ratio.

## 1. Introduction

Uric acid is a terminal product of purine degradation, and high levels of circulating uric acid have been associated with gout, type 2 diabetes (T2D), and renal and cardiovascular diseases [1]. Recently, a higher serum uric acid (SUA) to creatinine ratio was associated with an increased risk of metabolic syndrome [2], and predicted chronic kidney disease incidence in T2D patients [3] and chronic obstructive pulmonary disease as compared to uric acid alone [4]. Growing evidence also suggests that elevated SUA levels are frequently identified in patients with obstructive sleep apnea [5], possibly due to elevations in catecholamine levels [6] and intermittent hypoxia. Loss of sleep may also contribute to raised nocturnal catecholamine levels [7].

Sleep is a crucial determinant for metabolic homeostasis, and loss of sleep or disruptions of sleep–wake patterns have been associated with metabolic impairments. Sleep deprivation and sleep–wake cycle disturbances have been shown to activate proteolytic pathways which may affect the balance between protein synthesis and degradation, favoring catabolism [8,9]. As a result, proteins are broken down into their by-products, such as purines, which are metabolized to uric acid. To date, only one study without significant results has examined associations between subjective sleep duration and SUA levels [10].

No previous epidemiologic study has examined the association of objectively measured sleep characteristics with SUA and uric acid to creatinine ratio. Therefore, in the present cross-sectional study nested in the framework of the PREDIMED-Plus trial we tested the following two hypotheses: (1) shorter sleep duration is associated with higher SUA concentrations and higher SUA to creatinine ratio; and (2) higher sleep variability is associated with higher SUA concentrations and higher SUA to creatinine ratio.

## 2. Methods

### 2.1. Study Design and Population

We cross-sectionally analyzed data from the PREDIMED-Plus trial, a 6-year parallel-group, multicenter lifestyle intervention study involving 6874 participants recruited in 23 Spanish recruiting centers. Our analysis was performed before the implementation of the energy-restricted diet. The design of the PREDIMED-Plus trial has been described in detail elsewhere [11]. Community-dwelling adults (aged 55–75 years) with body mass index (BMI) ≥27 and <40 kg/m^2^, and meeting ≥3 metabolic syndrome individual components were included [12]. As previous studies have shown, females have a higher life expectancy than males, and since the main outcomes of PREDIMED-Plus study are cardiovascular diseases and mortality, we decided to recruit females aged at least 60 years and males aged equal or more than 55 in order to account for sex differences in life expectancy. We included participants with a BMI above 27 kg/m^2^ following the classification used by the Nutrition Screening Initiative [13] and adopted by Lipschitz [14], in which seniors with a BMI above 27 kg/m^2^ are classified as overweight. Out of the 6874 participants, data derived from accelerometry were available in a subsample of participants (*n* = 1993). Seven participants were excluded due to incomplete sleep data, resulting in a sample size of 1986 (Figure 1). Of these, 1842 participants had available uric acid measures (Figure 1). All participants provided written informed consent, and the study protocol and procedures were approved according to the ethical standards of the Declaration of Helsinki by all the participating institutions: El Comité de Ética de la Investigación (CEI) Provincial de Málaga, CEI de los Hospitales Universitarios Virgen Macarena y Virgen del Rocío, CEI de la Universidad de Navarra, CEI de las Illes Balears, El Comitè d’Ètica d’Investigació Clínica (CEIC) del Hospital Clínic de Barcelona, CEIC del Parc de Salut Mar, CEIC del Hospital Universitari Sant Joan de Reus, CEI del Hospital Universitario San Cecilio, CEIC de la Fundación Jiménez Díaz, CEIC Euskadi, CEI en Humanos de la Universidad de Valencia, CEIC del Hospital Universitario de Gran Canaria Doctor Negrín, CEIC del Hospital Universitario de Bellvitge, CEI de Córdoba, CEI de Instituto Madrileño De Estudios Avanzados, CEIC del Hospital Clínico San Carlos, CEI Provincial de Málaga, CEI de las Illes Balears, El Comité Coordinador de Ética d’Investigació (CCEI) de la Investigación Biomédica de Andalucía, and CEIC de León.

### 2.2. Biochemical Analyses

Fasting blood samples were collected, and laboratory technicians, who were blinded to the intervention group, performed biochemical analyses on fasting SUA using standard enzymatic procedures. Serum creatinine was measured by enzymatic reaction using the Jaffe method. Leukocyte counts were measured using an automated analyzer.

### 2.3. Sleep Assessment by Accelerometry

Participants were asked to wear a wrist-worn triaxial accelerometer (GENEActiv, ActivInsights Ltd, Kimbolton, United Kingdom) on their nondominant wrist for 8 consecutive days. Detailed methods of accelerometer data analysis are reported elsewhere [15]. The average nocturnal sleep duration, and intra-subject standard deviation of the sleep duration were calculated. 

### 2.4. Covariate Assessment

Information about age, sex, education, marital and employment status, smoking habits, depression, sleep apnea, and use of sedatives was provided by structured interviews. Adherence to an energy-restricted Mediterranean diet (MedDiet) was assessed using a 17-item questionnaire. A systolic and/or diastolic blood pressure ≥130/85 mmHg or antihypertensive drug usage were used to define high blood pressure. Moderate to vigorous physical activity (MVPA) was calculated and a new binary variable was created according to compliance of the WHO recommendations for MVPA set in ≥150 min/week [16]. T2D was defined as previous clinical diagnosis of diabetes, hemoglobin A1C (HbA1c) ≥ 6.5%, use of antidiabetic medication, or fasting plasma glucose >126 mg/dL.

### 2.5. Statistical Analysis

The normal distribution of the variables was evaluated using the Kolmogorov–Smirnov test. Participants’ characteristics according to categories of sleep duration are presented as means (±SD) or median (interquartile range) for quantitative variables, and percentages (%) and numbers (*n*) for categorical variables. These characteristics were examined across five categories of night sleep duration (<6 h, 6 to <7 h, 7 to <8 h, 8 to <9 h, and ≥9 h) [17]. One-way ANOVA (Bonferroni post hoc analysis for pairwise comparisons) or Kruskal–Wallis (Mann–Whitney tests in the post-hoc multiple comparisons) and chi-squared tests were used, as appropriate, to examine between-categories differences. Linear regression models were fitted to examine the associations of 1 h/day increment in sleep duration and sleep variability with SUA and SUA to creatinine ratio. A number of models were examined. For both outcomes, Model 1 was adjusted for sex and age (continuous). Model 2 was further adjusted for BMI (continuous), marital status (single/divorced, married, widower), employment (working, nonworking, retired), education (primary education, secondary education, academic/graduate), smoking habit (current smoker, past smoker, never smoked), sedative treatment (yes/no), sleep apnea (yes/no), T2D (yes/no), uric acid agents (allopurinol, febuxostat) (yes/no), hypertension (yes/no), depression (yes/no), 17-item energy-restricted Mediterranean diet (continuous), compliance with MVPA recommendations set at ≥150 min/week (yes/no), time spent in sustained inactivity bouts (“daytime napping”, min/day), and intervention center. A third model was fitted by including variables from Model 2 plus leukocytes in order to account for possible influence of this indirect inflammatory marker in the aforementioned associations [18,19,20]. The variables (sleep variability and daytime napping) that did not present a normal pattern were transformed logarithmically prior to these analyses. We used multiple imputation methods using the Stata “MI” module (the number of imputation was set to 20) to replace the missing values of leukocytes in 42 participants. Significance was set at *p*-values < 0.05.

## 3. Results

General characteristics as well as sleep and biochemical parameters for the entire population and by categories of nocturnal sleep duration are displayed in Table 1 and Table 2. The mean age of participants was 65.0 ± 5 years and mean nocturnal sleep duration was 6.9 ± 1.1 h, while the median sleep variability was 0.8 0.6; 1.1h, According to categories of sleep duration, those participants sleeping less than 6 hours were more likely to be younger and male compared to all the other sleep duration categories. They were also more likely to have a higher BMI, adhere less to an energy-restricted MedDiet, have higher education, and be currently employed. Regarding SUA and SUA to creatinine ratio, their values were higher in those participants sleeping less than 6 h. Spearman correlation coefficients among uric acid, creatinine, and leukocytes are shown in Appendix A. SUA was positively correlated with leukocytes (r = 0.11, *p* < 0.001).

In the fully adjusted model, a 1 hour/night increment in sleep duration was inversely associated with SUA concentrations (β = −0.07, *p* = 0.047) (Table 3). Further adjustment for leukocytes attenuated this association (*p* = 0.050). Furthermore, each 1-hour increment in sleep duration was inversely associated with the ratio of SUA to creatinine (β = −0.15, *p* = 0.001) and remained significant even after adjusting for leukocytes.

## 4. Discussion

In the present cross-sectional study of 1842 elderly participants from the PREDIMED-Plus trial, we observed inverse associations of sleep duration with SUA concentrations and the ratio of SUA to creatinine.

Elevations in catecholamine levels have been reported in participants with sleep loss [7], which may increase nucleotide turnover, thus enhancing endogenous uric acid production [21]. Furthermore, sleep deprivation may activate proteolytic pathways [8,9], leading to the production of purines and uric acid. Our results also suggest that the association between sleep duration and uric acid may be mediated by inflammation. Sleep deprivation may worsen systemic inflammation, as shown in two experimental studies in which the white blood cell and neutrophil counts were significantly higher after sleep restriction [18,19], leading to an increase in serum uric acid concentrations. Given the cross-sectional nature of this study, we cannot exclude the possibility that raised uric acid concentrations may cause inflammation by increasing leukocytes [20]. A previous study has observed that when levels of cytokines were compared between insomniacs and normal sleepers, there was a significant increase of cytokines in those with low sleep quality [22]. Another study found that chronic inflammation from any cause can lead to insomnia [23]. Whether both SUA and sleep loss can increase inflammatory markers requires further investigation. The increase in SUA to creatinine ratio in short sleepers may reflect the uric acid loading and/or a reduction in creatinine levels in circulation due to increased diuresis [24].

The current findings should be interpreted on the basis of the study’s limitations. First, the cross-sectional design does not allow any causal inference of the observed associations to be made. Second, the participants were elderly Mediterranean individuals with metabolic syndrome, thus, results cannot be extrapolated to other populations. Third, even though we adjusted for several potential confounders, residual confounding may remain. Fourth, we cannot exclude accuracy issues related to the derived sleep data. Using actigraphy in concert with complementary subjective methods such as sleep diaries may reduce these uncertainties.

In conclusion, “cross-sectional” evidence is useful to support the hypothesis that longer sleep duration is associated with lower SUA concentrations and lower SUA to creatinine ratio in elderly participants at high cardiovascular risk. Further longitudinal studies are required to clarify the temporal nature of this relationship and elucidate possible mechanisms underlying these observations.

## Figures and Tables

**Figure 1 nutrients-11-00761-f001:**
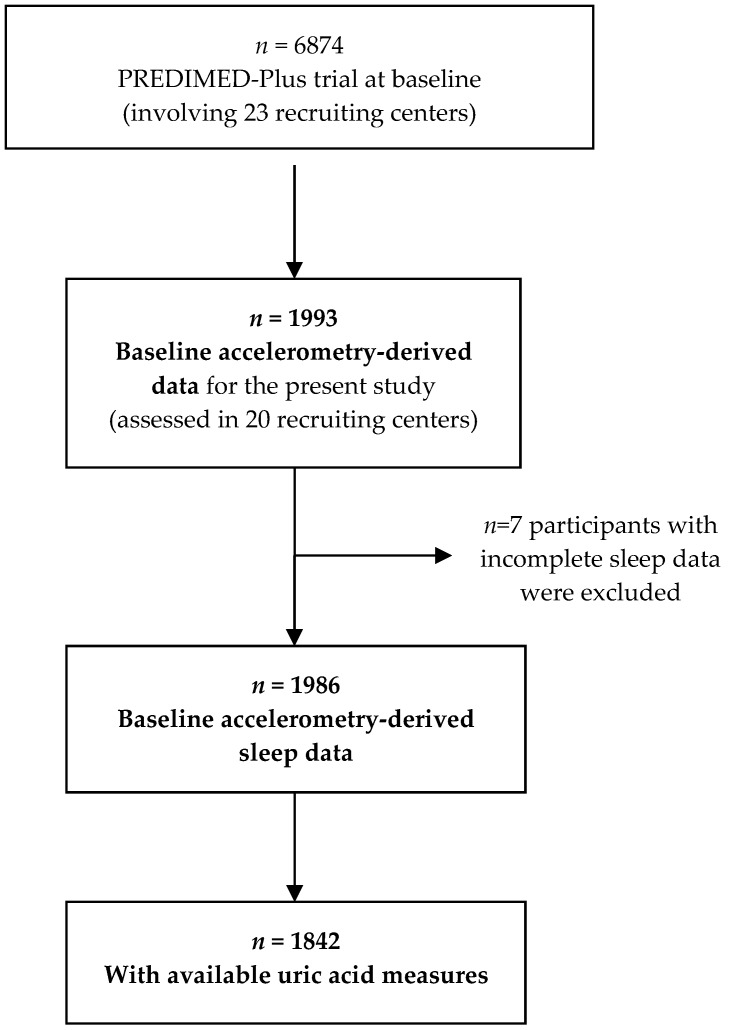
Flow-chart of study participants.

**Table 1 nutrients-11-00761-t001:** General characteristics of the study population from Prevención con Dieta Mediterránea-Plus trial across categories of nocturnal sleep duration.

			Categories of Nocturnal Sleep Duration (h)					
		Total *n* = 1842	<6 h *n* = 308	6–<7 h *n* = 618	7–<8 h *n* = 654	8–<9 h *n* = 225	≥9 h *n* = 37	*p*-Value
**Age, mean ± SD, years**		65 ± 5	64 ± 5	65 ± 5	65 ± 5	66 ± 4	68 ± 4	<0.001
**Male, *n* (%)**		979(53)	229(74)	348(56)	305(46)	85(38)	12(32)	<0.001
**BMI, mean ± SD, kg/m^2^**		32.6 ± 3.4	33.1 ± 3.5	32.5 ± 3.5	32.3 ± 3.4	32.6 ± 3.2	32.5 ± 3.4	0.031
**Type 2 diabetes, *n* (%)**		505(27)	95(31)	174(28)	170(26)	51(22)	15(40)	0.156
**Sleep apnea, *n* (%)**		237(13)	49(16)	73(12)	89(13)	23(10)	3(8)	0.229
**Depression, *n* (%)**		389(21)	59(19)	113(17)	153(23)	65(29)	11(30)	0.001
**Sedative treatment, *n* (%)**		442(24)	53(17)	114(18)	175(27)	86(38)	14(38)	<0.001
**Uric acid agents, *n* (%)**		168(9)	27(9)	62(10)	66(10)	10(4)	3(8)	0.117
**Smoking, *n* (%)**								
	Never	795(43)	91(29)	254(41)	314(48)	114(51)	22(61)	<0.001
	Former	828(45)	168(55)	289(47)	280(43)	82(36)	9(25)	
	Current	212(11)	48(15)	73(12)	58(9)	28(12)	5(14)	
**Adherence to energy-restricted MedDiet** **(score from 0 to 17 items), mean ± SD**		8.6 ± 2.7	8.2 ± 2.7	8.7 ± 2.7	8.5 ± 2.7	8.7 ± 2.7	9.6 ± 2.9	0.017
**Compliance of MVPA recommendations ^a^, *n* (%)**		352(19)	51(16)	133(21)	134(20)	33(13)	4(11)	0.027
**Education status, *n* (%)**								
	Primary education	909(49)	136(44)	278(45)	338(52)	132(59)	25(67)	<0.001
	Secondary education	494(27)	76(25)	175(28)	185(28)	52(23)	6(16)	
	Academic/graduate	413(22)	92(30)	157(25)	124(18)	35(15)	5(13)	
**Employment status, *n* (%)**								
	Working	376(20)	92(30)	148(24)	110(17)	25(11)	1(2)	<0.001
	Non-working	441(25)	62(18)	147(23)	182(26)	71(29)	10(26)	
	Retired	1017(55)	160(53)	331(54)	363(55)	136(60)	27(73)	
**Marital status, *n* (%)**								
	Single/divorced	267(14)	49(16)	95(15)	93(14)	22(10)	8(21)	0.029
	Married	1369(75)	227(74)	471(77)	477(73)	170(76)	24(65)	
	Widower	196(11)	31(10)	46(7)	82(12)	32(14)	5(13)	

Data is presented as mean ± SD unless otherwise indicated. Abbreviations: BMI, body mass index; MedDiet, Mediterranean Diet; MVPA, moderate to vigorous physical activity. ^a^ Recommendations for MVPA set at ≥150 min/week for elderly persons, based on accelerometry-derived 10-min bout MVPA.The *p*-value for differences between categories of nocturnal sleep duration was calculated by chi-squared or one-way analysis of variance test using the Bonferroni rule to correct for type I error in the post-hoc multiple comparisons for categorical and continuous variables, respectively.

**Table 2 nutrients-11-00761-t002:** Sleep and biochemical parameters of the study population from Prevención con Dieta Mediterránea-Plus trial across categories of nocturnal sleep duration.

			Categories of Nocturnal Sleep Duration (h)					
		Total *n* = 1842	<6 h *n* = 308	6–<7 h *n* = 618	7–<8 h *n* = 654	8–<9 h *n* = 225	≥9 h *n* = 37	*p*-Value
**Sleep parameters**								
	Sleep duration, mean ± SD, h	6.9 ± 1.1	5.3 ± 0.7	6.5 ± 0.3	7.4 ± 0.3	8.4 ± 0.2	9.3 ± 0.3	<0.001
	Sleep variability, median interquartile range, h	0.8 0.6; 1.1	0.90.6; 1.2	0.8 0.6; 1.0	0.8 0.6; 1.1	0.7 0.6; 1.0	0.8 0.6; 0.9	<0.001
	Napping duration, median interquartile range, min	76.749.7; 109.8	64.842.3; 91.1	67.742.9; 97.1	78.8 53.5; 108.6	113.4 79.9; 145.2	163.9 109.8; 203.5	<0.001
**Biochemical parameters**								
Serum uric acid, mean ± SD, mg/dL		6.0 ± 1.4	6.3 ± 1.5	6.1 ± 1.5	5.9 ± 1.4	5.7 ± 1.4	5.4 ± 1.5	<0.001
Creatinine, mean ± SD, mg/dL		0.83 ± 0.2	0.85 ± 0.2	0.82 ± 0.2	0.83 ± 0.2	0.83 ± 0.2	0.80 ± 0.2	0.165
Serum uric acid to creatinine ratio		7.4 ± 1.9	7.6 ± 1.9	7.7 ± 1.9	7.4 ± 1.8	7.1 ± 1.8	6.9 ± 2.2	<0.001
Leukocytes, mean ± SD, counts		6.7 ± 1.7	6.9 ± 1.8	6.7 ± 1.7	6.5 ± 1.6	6.8 ± 1.8	6.7 ± 1.4	0.090

Data is presented as mean ± SD or median interquartile range. The *p*-value for differences between categories of nocturnal sleep duration was calculated by chi-squared or one-way analysis of variance test using the Bonferroni rule to correct for type I error in the post-hoc multiple comparisons for categorical and continuous variables, respectively. In case of non-normally distributed variables we performed Kruskal–Wallis test with Mann–Whitney tests in the post-hoc multiple comparisons.

**Table 3 nutrients-11-00761-t003:** β-coefficients (95% Confidence Interval (CI)) for association between serum uric acid, serum uric acid to creatinine ratio, and sleep measures (sleep duration, sleep variability (log-transformed)).

	Model 1	*p*-Value	Model 2	*p*-Value	Model 3	*p*-Value
**SUA**						
Sleep duration (h)	−0.08 (−0.14, −0.02)	0.008	−0.07 (−0.13, −0.01)	0.047	−0.06 (−0.13, 0.01)	0.050
Sleep variability (h)	−0.29 (−0.60, 0.01)	0.056	−0.26 (−0.56, 0.05)	0.100	−0.25 (−0.56, 0.05)	0.101
**SUA/Cr**						
Sleep duration (h)	−0.20 (−0.29, −0.12)	<0.001	−0.15 (−0.24, −0.06)	0.001	−0.15 (−0.24, −0.06)	0.001
Sleep variability (h)	−0.30 (−0.72, 0.12)	0.163	−0.23 (−0.66, 0.19)	0.279	−0.24 (−0.67, 0.19)	0.272

Abbreviations: SUA, serum uric acid; Cr, creatinine. Model 1 adjusted for sex, age (years). Model 2 adjusted for Model 1 plus body mass index (kg/m^2^), marital status (single/divorced, married, widower), employment (working, nonworking, retired), education (primary education, secondary education, academic/graduate), smoking habit (current smoker, past smoker, never smoked), sedative treatment (yes/no), sleep apnea (yes/no), type 2 diabetes (yes/no), uric acid agents (yes/no), hypertension (yes/no), depression (yes/no), 17-item energy-restricted Mediterranean diet, compliance to MVPA recommendations set at ≥150 min/week (yes/no), time spent in sustained inactivity bouts (“daytime napping”, min/day) and intervention center. Model 3 adjusted for Model 2 plus leukocytes.

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
