# Peer review of "Sleep Duration is Inversely Associated with Serum Uric Acid Concentrations and Uric Acid to Creatinine Ratio in an Elderly Mediterranean Population at High Cardiovascular Risk"

_nutrients, 2019, doi:10.3390/nu11040761_

Reviewer 1 Report

This manuscript examines the associations between serum uric acid and sleep duration and variability. The manuscript is well written and examines a novel area. I have several comments/questions:

Abstract: Please add how sleep duration and variability were measured.

Introduction: a statement of hypotheses should be included.

Line 125: how is the MedDiet questionnaire scored? Is this used as a continuous variable?

Line 133: which post-hoc test was used for the ANOVA data?

Line 148: please cite work that implicates leukocytes as a moderating factor

Line 153: please provide the SD for age

Line 154: you say that participants sleeping less than 6 hours were more likely….is this compared to all other sleep categories? More detail is necessary, especially when the table does not provide this information. Please also explain how models 2 and 3 fared for both SUA and SUA/Cr in the text.

Discussion line 14 – please provide a p-value. Was this information included in the results section? It should be.

Line 24: does the energy-restricted diet bias your results in favor or against your hypotheses.

What’s the take home message? Why should we care about these results?

Author Response

This manuscript examines the associations between serum uric acid and sleep duration and variability. The manuscript is well written and examines a novel area. I have several comments/questions:

Abstract: Please add how sleep duration and variability were measured.

Reply: We have previously written that “Accelerometry-derived sleep duration and sleep variability were registered”. Therefore, sleep duration and variability were measured using accelerometers.

Introduction: a statement of hypotheses should be included.

Reply: Following this comment we added a statement of hypotheses: “Therefore, in the present cross-sectional study nested in the framework of the PREDIMED-Plus trial we tested the following two hypotheses: (1) shorter sleep duration is associated with higher SUA concentrations and higher SUA to creatinine ratio; (2) higher sleep variability is associated with higher SUA concentrations and higher SUA to creatinine ratio”.

Line 125: how is the MedDiet questionnaire scored? Is this used as a continuous variable?

Reply: The MedDiet questionnaire score ranges from 0 to 17. This variable was used as a continuous variable in the current analysis.

Line 133: which post-hoc test was used for the ANOVA data?

Reply: We used a Bonferroni post hoc analysis for pairwise comparisons. In case of non-normally distributed variables we performed Kruskal-Wallis test with Mann-Whitney tests in the post-hoc multiple comparisons. We added these in Statistical Analysis part and in the footnote of Table 1.

Line 148: please cite work that implicates leukocytes as a moderating factor

Reply: We cited with 3 References: J Thromb Thrombolysis. 2009;28(1):46-49.; Vasc Health Risk Manag. 2008;4(6):1467-1470.; Eur Heart J. 2006;27(10):1174-1181.

Line 153: please provide the SD for age

Reply: We added SD.

Line 154: you say that participants sleeping less than 6 hours were more likely….is this compared to all other sleep categories? More detail is necessary, especially when the table does not provide this information. Please also explain how models 2 and 3 fared for both SUA and SUA/Cr in the text.

Reply: Participants sleeping less than 6 hours were more likely to be younger and male compared to all other sleep categories. We provided this information in the relevant sentence in the text. We have explained in statistical part that models 2 and 3 were used for both SUA and SUA/Cr.

Discussion line 14 – please provide a p-value. Was this information included in the results section? It should be.

Reply: We provided a p-value and following the Reviewer’s comment we moved this sentence to Results section.

 Line 24: does the energy-restricted diet bias your results in favor or against your hypotheses.

Reply: Our analysis is cross-sectional before the implementation of the energy-restricted diet. Therefore, the diet does not affect our results.

What’s the take home message? Why should we care about these results?

Reply: As we wrote in Discussion section: “we observed inverse associations of sleep duration with SUA concentrations and the ratio of SUA to creatinine.” Therefore, this "cross-sectional" evidence is useful to support the hypothesis that longer sleep duration is associated with lower uric acid concentrations and the aforementioned ratio. We modified the conclusion part accordingly.

Reviewer 2 Report

The manuscript entitled “Sleep duration is inversely associated with serum uric acid concentrations and uric acid to creatinine ratio in an elderly Mediterranean population at high cardiovascular risk” presents interesting issue, but it requires some corrections before being published.

One major ethical problem is associated with the authorship. It is understandable that in case of a large study there are a number of authors (there are 52 Authors of the prepared manuscript), but all of them must participate actively in the manuscript preparation. All Authors must at least participate in critical review of the manuscript (if not participate in writing it).

1.      Based on Authors Contributions section, in fact there are only 2 Authors who prepared the manuscript (as specified – CP and JSS analysed the data and wrote the manuscript). But majority of other Authors only participated in data gathering. In such situation, if they did not participate in manuscript preparation, they should not be indicated as authors, but rather mentioned in Acknowledgements section.

2.      Based on Authors Contributions section, there are some Authors (e.g. NBT) who did nothing for the study and did nothing for the manuscript preparation (is not indicated in Authors Contributions section at all). Such persons should be not indicated as Author of the study

Authors should remember, that the “guest authorship” is the forbidden procedure. Authors who did not participate in the manuscript preparing should be removed at all or just be presented in Acknowledgements Section. If they participated actively in the study and creating manuscript, as well as performed critical revisions of the manuscript and accepted the publication of the data, they should be indicated, but their contributions should be clearly specified.

General:

1.      The manuscript is in general shabbily prepared, as there are e.g. typos, lacking spaces, additional symbols where they are not needed, lack of superscript where it should be.

2.      Moreover, manuscript was not prepared according to the instructions for authors (e.g. numbering of chapters, not continuous line numbering, References section).

3.      Authors must precisely formulate their observations, in order to present what exactly is being observed, e.g. instead of “sleep duration is associated with lower SUA concentrations”, it should be either “longer sleep duration is associated with lower SUA concentrations”, or “sleep duration is associated with SUA concentrations”, or “sleep duration is inversely associated with SUA concentrations”

Abstract:

Instead of what was done, Authors should specify the aim (e.g. “The aim of the study was…”).

Introduction:

Authors properly justified the study.

Materials and Methods:

Authors did not describe the conducted study. They referred “Response to letter to the editor from Dr Rahman Shiri: The challenging topic of suicide across occupational groups” while stating tat there is the PREDIMED-Plus trial presented, but it is not. They should precisely describe the most important issues in their manuscript.

Authors should justify the included age groups, as for male and female participants the included age range differed.

Authors should justify the BMI criteria, as BMI of 27 kg/m2 is not in agreement with general criteria by WHO.

Supplementary Figure 1 – should be included to the main body of the study.

Authors should justify the decided categories of sleep duration (e.g. reference is needed).

It seems, that Authors did not verify the normality of distribution for the assessed variables. Authors must verify the normality of distribution and specify the test applied for verification.

If the distribution is normal, the mean values should be presented (accompanied by SD), but if it is different than normal, the median, accompanied by minimum and maximum values should be presented – it should be specified that distribution is normal if it is. 

The applied test should be chosen taking into account the observed distribution.

Results:

It seems, that Authors did not verify the normality of distribution for the assessed variables. Authors must verify the normality of distribution and specify the test applied for verification.

If the distribution is normal, the mean values should be presented (accompanied by SD), but if it is different than normal, the median, accompanied by minimum and maximum values should be presented – it should be specified that distribution is normal if it is. 

The applied test should be chosen taking into account the observed distribution.

Table 1 – is hard to follow, as it is presented on 2 pages – Authors should think about dividing it into 2 separate ones

Discussion:

Authors should not reproduce the methodology and results of the study

Authors should present also other limitations of the study (e.g. associated with the sleep duration measurement).

Author Response

The manuscript entitled “Sleep duration is inversely associated with serum uric acid concentrations and uric acid to creatinine ratio in an elderly Mediterranean population at high cardiovascular risk” presents interesting issue, but it requires some corrections before being published.

One major ethical problem is associated with the authorship. It is understandable that in case of a large study there are a number of authors (there are 52 Authors of the prepared manuscript), but all of them must participate actively in the manuscript preparation. All Authors must at least participate in critical review of the manuscript (if not participate in writing it).

Reply: All authors have participated in critical review of the manuscript as indicated before: “All authors revised the manuscript for important intellectual content and read and approved the final manuscript.”

We have also modified the authorship statement to indicate that all authors participated actively in the study and deserve to be among the authors:

MA.M-G., D.C., D.R., J.V., AM.A-G., J.W., J.A.M., J.L-M., R.E., X.P., JA.T., A.B-C., M.D-R., P.M-M., L.D., V.M., J.V., C.V., E.R., J.S-S designed the PREDIMED-Plus study. C.P, N.B, J.S-S, designed the research. C.P., N.B., A.D-L., MA.M-G., N.B-T., D.C., H.S., M.F., J.V., D.R., J.V., AM.A-G., J.W., AJ.M., J.A.M., J.L-M., R.E., A.B-C., F.A., J.A.T., F.J.T., L.S-M., R.E., A.MG., V.M., J.L, C.V., X.P., JA.T., A.G-R., A.B-C., M.D-D., P.M-M., J.V., L.D., V.M.S., J.V., C.V., M.D-R., E.R., M.R-C.,I.A., J.B-L., A.G-A., M.B., JV.S., M.Q., A.C., A.O-C., L.T.S., J.B-L., N.P-F., I.A., A.S-V., R.C., J.C.F.G., J.M.S-L., E.C., M.M.B.,  J.D-E., EM.A., L.T., M.M., L.C-G., I.S.L., J.C.C-O., O.C., H.S., E.T., A.A-G., and J.S-S., D.C., M.F., J.V., D.R., J.A.M., J.L-M., R.E., A.B-C., F.A., J.A.T., F.J.T., L.S-M., V.M., J.L, C.V., X.P., J.V., L.D., M.D-R., E.R., I.A., J.B-L., A.G-A., M.B., H.S., E.T., A.A-G., and J.S-S., conducted the research. C.P, J.S-S, analyzed the data. C.P, J.S-S, wrote the article. C.P, J.S-S, are the guarantors of this work and, as such, had full access to all the data in the study and take responsibility for the integrity of the data and the accuracy of the data analysis. All authors revised the manuscript for important intellectual content and read and approved the final manuscript.

1.      Based on Authors Contributions section, in fact there are only 2 Authors who prepared the manuscript (as specified – CP and JSS analysed the data and wrote the manuscript). But majority of other Authors only participated in data gathering. In such situation, if they did not participate in manuscript preparation, they should not be indicated as authors, but rather mentioned in Acknowledgements section.

Reply: We have modified the authorship statement to indicate that all authors participated actively in the study and critical review of the manuscript and deserve to be among the authors (Please see the previous answer).

2.      Based on Authors Contributions section, there are some Authors (e.g. NBT) who did nothing for the study and did nothing for the manuscript preparation (is not indicated in Authors Contributions section at all). Such persons should be not indicated as Author of the study

Reply: We have modified the authorship statement to indicate that all authors, including NBT, participated actively in the study and critical review of the manuscript and deserve to be among the authors (Please see the previous answer).

Authors should remember that the “guest authorship” is the forbidden procedure. Authors who did not participate in the manuscript preparing should be removed at all or just be presented in Acknowledgements Section. If they participated actively in the study and creating manuscript, as well as performed critical revisions of the manuscript and accepted the publication of the data, they should be indicated, but their contributions should be clearly specified.

Reply: We have modified the authorship statement to indicate that all authors, including NBT, participated actively in the study and critical review of the manuscript and deserve to be among the authors (Please see the previous answer).

General:

1.      The manuscript is in general shabbily prepared, as there are e.g. typos, lacking spaces, additional symbols where they are not needed, lack of superscript where it should be.

Reply: We followed this comment and we edited the manuscript.

2.      Moreover, manuscript was not prepared according to the instructions for authors (e.g. numbering of chapters, not continuous line numbering, References section).

Reply: We have numbered pages and deleted line numbering. Regarding references, we have followed the Instructions for Authors:

In the text, reference numbers should be placed in square brackets [ ], and placed before the punctuation; for example [1], [1–3] or [1,3]. 

Journal Articles:
1. Author 1, A.B.; Author 2, C.D. Title of the article. Abbreviated Journal Name YearVolume, page range. Available online: URL (accessed on Day Month Year).

3.      Authors must precisely formulate their observations, in order to present what exactly is being observed, e.g. instead of “sleep duration is associated with lower SUA concentrations”, it should be either “longer sleep duration is associated with lower SUA concentrations”, or “sleep duration is associated with SUA concentrations”, or “sleep duration is inversely associated with SUA concentrations”

Reply: We fully agree with the reviewer and we modified all relevant phrases through the manuscript.

Abstract:

Instead of what was done, Authors should specify the aim (e.g. “The aim of the study was…”).

Reply: We changed it accordingly.

Introduction:

Authors properly justified the study.

Reply: We would like to thank the Reviewer.

Materials and Methods:

Authors did not describe the conducted study. They referred “Response to letter to the editor from Dr Rahman Shiri: The challenging topic of suicide across occupational groups” while stating tat there is the PREDIMED-Plus trial presented, but it is not. They should precisely describe the most important issues in their manuscript.

Reply: By mistake we cited the above response to letter to the editor instead of the paper entitled “Martínez-González MABuil-Cosiales PCorella D, Bulló M, Fitó M, Vioque J, et al. Cohort Profile: Design and methods of the PREDIMED-Plus randomized trial. Int J Epidemiol. 2018. doi: 10.1093/ije/dyy225”. We have corrected it.

Authors should justify the included age groups, as for male and female participants the included age range differed.

Reply: As previous studies have shown, females have a higher life expectancy than males and since the main outcomes of PREDIMED-Plus study are cardiovascular diseases and mortality we decided to recruit females with a starting age point of 60 and males with a starting age of 55 in order to account for sex differences in life expectancy.

Authors should justify the BMI criteria, as BMI of 27 kg/m2 is not in agreement with general criteria by WHO.

Reply: In the 1990s, another classification based on BMI was proposed for the diagnosis of obesity, adapted to the elderly population. This classifi-cation is used by the Nutrition Screening Initiative (NSI)[1] and adopted by Lipschitz[2], in which seniors with a BMI above 27kg/m2 are classified as overweight, while those with a BMI below 22kg/m2 are classified as thin. Lipschitz[2] says that the use of these values from the lesser mortality of the elderly in this BMI range, however, does not refer to changes of aging.

1.      Nutrition Screening Initiative. Nutrition interventions manual for professionals caring for older Americans. Washington DC: Nutrition Screening Initiative; 1992.

2.      Lipschitz DA. Screening for nutritional status in the elderly. Prim Care 1994; 21:55-67.

Supplementary Figure 1 – should be included to the main body of the study.

Reply: Following the Reviewer’s comment we included Supplementary Figure 1 in the main body of the study and renamed it to Figure 1.

Authors should justify the decided categories of sleep duration (e.g. reference is needed).

Reply: Categories of sleep duration were defined based on the 2015 National Sleep Foundation recommendations[1]. We cited it in the manuscript.

1.      Hirshkowitz M, Whiton K, Albert SM, et al. National Sleep Foundation’s updated sleep duration recommendations: Final report. Sleep Heal. 2015;1(4):233-243. doi:10.1016/j.sleh.2015.10.004.

It seems, that Authors did not verify the normality of distribution for the assessed variables. Authors must verify the normality of distribution and specify the test applied for verification.

Reply: We used Kolmogorov-Smirnov test to evaluate the normal distribution of the variables. All variables are normally distributed except for sleep variability and daytime napping.

If the distribution is normal, the mean values should be presented (accompanied by SD), but if it is different than normal, the median, accompanied by minimum and maximum values should be presented – it should be specified that distribution is normal if it is.

Reply: We replaced mean values and SD with median (interquartile range) for sleep variability and daytime napping because they are not normally distributed.

The applied test should be chosen taking into account the observed distribution.

Reply: The associations of 1 hour/day increment in sleep duration and sleep variability with SUA and SUA to creatinine ratio were examined using linear regression models with sleep variability and daytime napping log-transformed.

Results:

It seems, that Authors did not verify the normality of distribution for the assessed variables. Authors must verify the normality of distribution and specify the test applied for verification.

Reply: We used Kolmogorov-Smirnov test to evaluate the normal distribution of the variables. All variables are normally distributed except for sleep variability and daytime napping.

If the distribution is normal, the mean values should be presented (accompanied by SD), but if it is different than normal, the median, accompanied by minimum and maximum values should be presented – it should be specified that distribution is normal if it is.

Reply: We replaced mean values and SD with median (interquartile range) for sleep variability and daytime napping because they are not normally distributed.

The applied test should be chosen taking into account the observed distribution.

Reply: The associations of 1 hour/day increment in sleep duration and sleep variability with SUA and SUA to creatinine ratio were examined using linear regression models with sleep variability and daytime napping log-transformed.

Table 1 – is hard to follow, as it is presented on 2 pages – Authors should think about dividing it into 2 separate ones

Reply: We divided Table 1 into 2 separate ones in order to be easier to follow.

Discussion:

Authors should not reproduce the methodology and results of the study

Reply: We followed this comment and rephrased “After adjusting the analyses for leukocytes count, our results changed notably suggesting that the association between sleep duration and uric acid may be mediated by inflammation.” to “Our results also suggest that the association between sleep duration and uric acid may be mediated by inflammation.” Furthermore, we deleted “To strengthen our hypothesis, Spearman's correlation analysis between SUA and leukocytes showed a significantly positive correlation (r = 0.11).” and moved it to Results section.

Authors should present also other limitations of the study (e.g. associated with the sleep duration measurement).

Reply: The use of accelerometry is a reliable method to assess sleep data as compared to other subjective tools and therefore is strength of our study. However, there are limitations to the accuracy of the derived sleep data. Periods of low activity in which patients lay quietly in bed but are awake may be scored as sleep by actigraphy software. When performing actigraphy, it is essential to require participants to complete a sleep diary which was not available in our study. We added the following sentence in limitations part: “Fourth, we cannot exclude accuracy issues related to the derived sleep data. Using actigraphy in concert with complementary subjective methods such as sleep diaries may reduce these uncertainties”.

Round  2

Reviewer 1 Report

Abstract: "registered" is not the correct word. Use "measured" instead.

Please state that the MedDiet score was a continuous variable.

Explain that the energy restriction occurred after the measurements.

Author Response

Abstract: "registered" is not the correct word. Use "measured" instead.

Reply: Following this comment we changed “registered” to “measured”.

Please state that the MedDiet score was a continuous variable.

Reply: We made it clear in statistical part.

Explain that the energy restriction occurred after the measurements.

Reply: We added in study design and population the following sentence: “Our analysis was performed before the implementation of the energy-restricted diet.

Reviewer 2 Report

The manuscript entitled “Sleep duration is inversely associated with serum uric acid concentrations and uric acid to creatinine ratio in an elderly Mediterranean population at high cardiovascular risk” presents interesting issue, but it still requires some corrections before being published, as some issues were still not corrected.

General:

1.      Some parts of the manuscript are still shabbily prepared (e.g. References section).

2.      Moreover, manuscript was not prepared according to the instructions for authors (e.g. numbering of chapters, line numbering, References section).

Abstract:

Authors should briefly justify the study in this section.

Materials and Methods:

Authors should justify the included age groups, as for male and female participants the included age range differed. They should justify it in their manuscript – not in response letter.

Authors should justify the BMI criteria, as BMI of 27 kg/m2 is not in agreement with general criteria by WHO. They should justify it in their manuscript – not in response letter.

Number of ethical commission agreement should be specified (referred).

Author Response

General:

1.      Some parts of the manuscript are still shabbily prepared (e.g. References section).

Reply: We modified References section according to the Instructions for Authors.

2.      Moreover, manuscript was not prepared according to the instructions for authors (e.g. numbering of chapters, line numbering, References section).

Reply: We have numbered pages and deleted line numbering. Regarding references, we have followed the Instructions for Authors

 Abstract:

Authors should briefly justify the study in this section.

Reply: In the previous round of revision the Reviewer suggested to delete what was done “Research examining the association of sleep characteristics with uric acid is lacking.” that could justify the study.

Instead of what was done, Authors should specify the aim (e.g. “The aim of the study was…”)”.

 Materials and Methods:

Authors should justify the included age groups, as for male and female participants the included age range differed. They should justify it in their manuscript – not in response letter.

Reply: We added the following sentence in the manuscript: “As previous studies have shown, females have a higher life expectancy than males and since the main outcomes of PREDIMED-Plus study are cardiovascular diseases and mortality we decided to recruit females with at least 60 years and males aged equal or more than 55 in order to account for sex differences in life expectancy.

Authors should justify the BMI criteria, as BMI of 27 kg/m2 is not in agreement with general criteria by WHO. They should justify it in their manuscript – not in response letter.

Reply: We added the following text:

We included participants with a BMI above 27kg/m2 following the classification used by the Nutrition Screening Initiative[13] and adopted by Lipschitz[14], in which seniors with a BMI above 27kg/m2 are classified as overweight.

Number of ethical commission agreement should be specified (referred)

Reply: We added the following paragraph:

All participants provided written informed consent, and the study protocol and procedures were approved according to the ethical standards of the Declaration of Helsinki by all the participating institutions: CEI Provincial de Málaga, CEI de los Hospitales Universitarios Virgen Macarena y Virgen del Rocío, CEI de la Universidad de Navarra, CEI de las Illes Balears, CEIC del Hospital Clínic de Barcelona, CEIC del Parc de Salut Mar, CEIC del Hospital Universitari Sant Joan de Reus, CEI del Hospital Universitario San Cecilio, CEIC de la Fundación Jiménez Díaz, CEIC Euskadi, CEI en Humanos de la Universidad de Valencia, CEIC del Hospital Universitario de Gran Canaria Doctor Negrín, CEIC del Hospital Universitario de Bellvitge, CEI de Córdoba, CEI de Instituto Madrileño De Estudios Avanzados, CEIC del Hospital Clínico San Carlos, CEI Provincial de Málaga, CEI de las Illes Balears, CCEI de la Investigación Biomédica de Andalucía and CEIC de León.